# The Effects of Habitual Foot Strike Patterns on the Morphology and Mechanical Function of the Medial Gastrocnemius–Achilles Tendon Unit

**DOI:** 10.3390/bioengineering10020264

**Published:** 2023-02-17

**Authors:** Lu Li, Kaicheng Wu, Liqin Deng, Cuixian Liu, Weijie Fu

**Affiliations:** School of Exercise and Health, Shanghai University of Sport, Shanghai 200438, China

**Keywords:** medial gastrocnemius, foot strike pattern, muscle morphology

## Abstract

As a crucial and vulnerable component of the lower extremities, the medial gastrocnemius–Achilles tendon unit (gMTU) plays a significant role in sport performance and injury prevention during long-distance running. However, how habitual foot strike patterns influence the morphology of the gMTU remains unclear. Therefore, this study aimed to explore the effects of two main foot strike patterns on the morphological and mechanical characteristics of the gMTU. Long-distance male runners with habitual forefoot (FFS group, n = 10) and rearfoot strike patterns (RFS group, n = 10) and male non-runners (NR group, n = 10) were recruited. A Terason uSmart 3300 ultrasonography system was used to image the medial gastrocnemius (MG) and Achilles tendon, Image J software to analyze the morphology, and a dynamometer to determine plantar flexion torque during maximal voluntary isometric contractions. The participants first performed a 5-minute warm up; then, the morphological measurements of MG and AT were recorded in a static condition; finally, the MVICs test was conducted to investigate the mechanical function of the gMTU. One-way ANOVA and nonparametric tests were used for data analysis. The significance level was set at a *p* value of <0.05. The muscle fascicle length (FL) (FFS: 67.3 ± 12.7, RFS: 62.5 ± 7.6, NRs: 55.9 ± 2.0, η2 = 0.187), normalized FL (FFS: 0.36 ± 0.48, RFS: 0.18 ± 0.03, NRs: 0.16 ± 0.01, η2 = 0.237), and pennation angle (PA) (FFS: 16.2 ± 1.9, RFS: 18.9 ± 2.8, NRs: 19.3 ± 2.4, η2 = 0.280) significantly differed between the groups. Specifically, the FL and normalized FL were longer in the FFS group than in the NR group (*p* < 0.05), while the PA was smaller in the FFS group than in the NR group (*p* < 0.05). **Conclusion:** Long-term running with a forefoot strike pattern could significantly affect the FL and PA of the MG. A forefoot strike pattern could lead to a longer FL and a smaller PA, indicating an FFS pattern could protect the MG from strain under repetitive high loads.

## 1. Introduction

Running is a popular exercise worldwide, as it is simple, inexpensive, time-efficient, and easily available [1]. Particularly, the popularity of long-distance running has increased approximately 345% over the last 10 years, and this figure continues to rise [2]. In running, the triceps surae muscle–tendon unit (MTU) is very important due to its unique function in generating force, as well as storing and releasing mechanical energy [3].

The morphological and mechanical properties of the MTU are related to running performance as well as injury occurrence [4,5,6,7]. Morphological features, such as the cross-sectional area (CSA) of the Achilles tendon (AT) and the muscle fascicle length (FL) related to the maximum shortening speed of the muscle [4], have been reported to strongly correlate with running performance [8,9]. Additionally, muscle strain also has been thought to relate to individual muscle morphology, which is known as an injury risk factor. A previous study showed that with a shorter FL of calf muscle, the risk of injury is higher [8]. A longer triceps surae FL could increase the ankle range of motion and reduce the predisposition for muscle strain, which may decrease the risk of injury [10].

The foot strike pattern has recently become a topic of academic interest and discussion among researchers, footwear manufacturers, and runners, in regard to the incidence of injury rates varying by landing pattern. Forefoot (FFS) and rearfoot (RFS) strike patterns are referred to differently, depending on whether the initial point of contact occurs close to the heel or around the ball of the foot. Kinematics and kinetics have been reported to significantly differ between these two patterns; for example, the FFS pattern yields a larger plantar flexion angle, larger knee flexion angle, a smaller knee moment during striking, and a greater plantarflexion moment during the stance phase compared with the RFS pattern [11,12]. The behavior of the MTU is expected to be affected, as this unit crosses both the knee and ankle joints, and plays an import role in lower extreme motor function. The medial gastrocnemius–Achilles tendon unit (gMTU), a crucial and vulnerable component of the MTU [13], is frequently used in studies as a representation of the morphological adaptation of the MTU [14]. However, because of the repetitive loading on the gMTU during running [3,15], whether adaptive changes in the gMTU occur after long-term running remains unclear. Furthermore, the effects of different habitual foot strike patterns and mechanical stimuli on the gMTU, especially on its morphological and mechanical properties, remain controversial. To date, little research has compared the morphological properties of muscle between FFS and RFS, and there is no consensus on the effect of long-term running on medial gastrocnemius (MG) and AT morphology. In a study by Abe et al. [16], the pennation angle (PA) of the MG was larger in long-distance runners than in the general population, while another study found no difference between long-distance runners and nonrunners (NRs) [17]. Rosager et al. reported that long-distance runners had a greater CSA than NRs [18], while Westh et al. found no significant difference in the normalized CSA between runners and NRs [19].

Therefore, this study aimed to investigate the effects of different foot strike patterns during running on the morphological and mechanical properties of the gMTU in vivo. Based on previous studies, we hypothesized that running habits could lead to changes both in morphological and mechanical features, such as a longer FL, larger PA, thicker muscle, and greater CSA and ankle plantar flexion torque. Moreover, the FFS runners would have a longer FL, larger PA, thicker muscle, and greater CSA and plantar flexion torque than RFS runners.

## 2. Materials and Methods

### 2.1. Participants

The required sample size was calculated as 30 (f = 0.63, α = 0.05, β = 0.2) using statistical software (G*Power version 3.1.9.6, Univ. Kiel, Kiel, Schleswig-Holstein, Germany). The effect size was calculated using data on the CSA of the AT previously reported by Intziegianni et al. [20]. Therefore, based on the result of the calculation, 10 male RFS runners, 10 male FFS runners, and 10 male NRs were recruited for the present study.

Before the formal measurements, all prospective participants completed a form to provide individual information, including basic body data, sport exercise condition, and recent medical information. Individuals who had sustained lower extremity injuries in the past 6 months and those who had a history of or current AT injuries, MG strains, or neurological disorders were excluded from the study. To ensure consistent and meaningful data, all RFS and FFS runners were habitual distance runners. This was defined as individuals who ran at least 15 km/week with their habitual foot strike pattern for the preceding 6 months [21]. NRs were defined as people who do not run or train regularly. Additionally, NRs should not meet the scoring criteria for being minimally active in the International Physical Activity Questionnaire Short Form [22]: (i) three or more days of vigorous activity lasting at least 20 min per day; (ii) five or more days of moderate-intensity activity or walking lasting at least 30 min per day; or (iii) five or more days of any combination of walking, moderate-intensity, or vigorous activities lasting at least 600 MET-minutes per week. Basic participant information is shown in Table 1. The participants were prohibited from performing any strenuous exercise for 24 h prior to the test to avoid fatigue affecting their performance. The study was approved by the institutional review board of the Shanghai University of Sport (number 102772021RT085), and all participants provided their informed consent before the official experiment.

### 2.2. Instrumentations

A B-mode ultrasonography system (Terason uSmart 3300, TeraTech, Burlington, MA, USA) with a linear array probe (12L5A; frequency: 7.5–12 MHz) was used to measure the morphological properties of the MG from the starting to the ending position. The CSA of the AT was measured using the same ultrasonography system. A dynamometer (CONTREX MJ, Physiomed, Schnaittach, Germany; sampling frequency: 256 Hz) was used to determine the plantar flexion torque during maximal voluntary isometric contractions (MVICs) to investigate the mechanical properties of the gMTU. Podoon© pressure-sensitive intelligent shoe pads (Paodong Inc., Xiamen, Fujian, China; 100 Hz, 3 mm thickness) were used to monitor the foot strike pattern [23,24]. By replacing the original insoles and linking them to a mobile application on tablet, this system allowed the foot strike pattern to be tracked in real time. To detect and recognize the correct foot strike pattern, the forefoot strike pattern was determined when only the sensor placed on the metatarsophalangeal joint was triggered at least two frames prior in each gait cycle, and the heel sensor was not triggered. The validity and reliability of the sensors and system were verified by the manufacturer.

### 2.3. Procedures

At the beginning of the experiment, the participants were asked to replace their shoes with the shoes provided by researchers. To avoid the interferences of different types of running shoes and to inspect the foot strike pattern, a particularly designed shoe, which had the original insole replaced with a pressure-sensitive smart insole (Nike Air Zoom Pegasus 34, USA; foam and air cushion midsole, 12 mm heel-to-toe drop), was implemented in present study. After a short adaptive phase, the participants were asked to run on the treadmill for a 5 min warm up at a speed of 12 km/h with their habitual foot strike pattern. The strike pattern was determined according to the real-time feedback from the smart insoles. After the warm-up period, the participants were asked to sit with their hip and knee flexed at 90° (0° = fully extended) and the ankle at 90° (the ankle was perpendicular to the shank) to measure individual morphological parameters. The shank length was measured from the medial tibial condyle to the medial malleolus of the ankle for standardization of the AT length (L*_AT_*) and FL during this phase [25].

Subsequently, the participants were instructed to lie prostrate on a treatment bed with their ankles in a neutral position with proper fixation (with the shank perpendicular to the foot and the ankle angle at 90°). One ultrasonography expert prepared the ultrasound system and the participants (including cleaning the participant’s skin and applying the ultrasonic coupling agent on the target position). Thereafter, the ultrasound probe was positioned perpendicular to the skin and longitudinal to the orientation of the muscle fibers at the MG belly (30% of the distance between the popliteal crease and the lateral malleolus) to record the morphological images of the MG [26]. The ultrasound probe was positioned parallel to the superficial and deep aponeuroses, with the aligned hyperechoic perimysial intramuscular connective tissue clearly visualized. After adjusting the ultrasound system settings to yield optimal-quality images, clear images of the MG were recorded at least 3 times for data analysis.

The L*_AT_* was defined as the distance between the insertion points of the AT at the calcaneus and junction point of the AT and MG, as determined on ultrasound [27] (Figure 1). The CSA of the AT on the horizontal position of the medial and lateral malleoli was also determined [28].

After the ultrasound measurement, participants were then asked to lie prone on the dynamometer with their ankle in the neutral position, knee fully extended, and thigh fixed. Participants practiced the MVICs test several times with guidance, avoiding peak performance. The participants then performed MVICs three times with encouragement. All tests were conducted by the same investigators. The flow chart for the process is shown in Figure 2.

### 2.4. Data Analysis

All ultrasound images were analyzed using Image J (NIH, Bethesda, MD, USA) [26,27,28].

(1) The FL was defined as the length of the fascicular path between the superficial and deep aponeuroses or as the length of the intersection of the muscle fascicle extension line and the aponeuroses extension line when the ends of the fascicles were outside of the ultrasound image [29]. It was measured as the mean value of the three fascicles of the best ultrasound image of each participant. The normalized FL was defined as the FL divided by the individual shank length [16].

(2) The PA was described as the angle between the muscle fascicle and the deep aponeuroses, and calculated as the mean value from three ultrasound images of each participant (Figure 3a) [26].

(3) The muscle thickness (MT) was defined as the distance between the deep and superficial aponeuroses, and was calculated as the mean value of five parallel lines drawn at right angles between the superficial and deep aponeuroses (Figure 3a) [26].

(4) The CSA of the AT was described as the surrounding echogenic boundary of the AT (Figure 3b) [30].

(5) The L*_AT_* was defined as the distance between the AT–soleus myotendinous junction and AT insertion. The normalized L*_AT_* was divided by the shank length.

(6) The peak plantar flexion torque (T_MAX_) of the ankle was measured using a dynamometer, and the normalized T_MAX_ was divided by the body weight.

Furthermore, to ensure the consistency and accuracy of the test results and avoid a manual measurement error, a corresponding reliability test was performed in the pre-experiment. The intraclass correlation coefficient (ICC) was measured by obtaining the PA, FL, and MT data to evaluate the reliability of the measurements of different experimenters. The results showed that the method and data analysis process of the present study had good reliability (ICC = 0.810–0.983).

**Figure 3 bioengineering-10-00264-f003:**
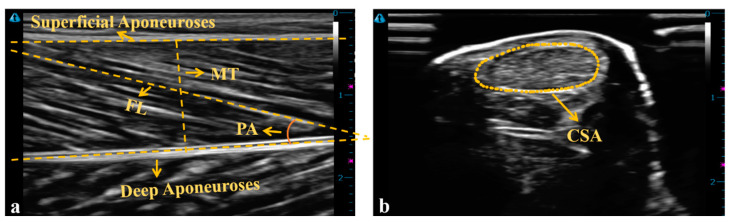
(**a**) The ultrasound image of MG. (**b**) The ultrasound image of the AT’s CSA.

### 2.5. Statistical Analysis

All parameters are presented as means ± standard deviations. Statistical analysis was performed using IBM SPSS version 26 (IBM Statistics, Chicago, IL, USA). The Shapiro–Wilk test was used to test the data normality; one-way ANOVA was performed for variables with a normal distribution, and nonparametric testing (Kruskal–Wallis test) was used for those with a non-normal distribution. The PA, FL, and normalized FL were non-normally distributed; thus, nonparametric tests were used to inspect the difference in these parameters between the groups. One-way ANOVA was used for the remaining parameters. The significance level was set at a *p* value of < 0.05. η2 was used to quantify the effect size.

## 3. Results

### 3.1. MG Morphology

The FL, normalized FL, and PA showed a significant difference between two groups (Table 2). The post hoc test showed that the FFS runners had a longer FL, longer normalized FL (*p* = 0.020, Figure 4), and smaller PA than the NRs. However, there were no significant differences found in the MT.

### 3.2. AT Morphology

No significant difference in the L*_AT_*, normalized L*_AT_*, and CSA was found between the groups.

### 3.3. Ankle Torque during MVICs

The T*_MAX_* and normalized T*_MAX_* also did not significantly differ between the three groups.

**Table 2 bioengineering-10-00264-t002:** The morphological differences in MG and AT under different habitual foot strike patterns.

Variable	FFS (n = 10)	RFS (n = 10)	NRs (n = 10)	*p* Value	*η* ^2^
shank length (cm)	33.7 ± 2.4	34.1 ± 2.1	34.9 ± 2.0	0.641	0.849
FL (mm)	67.3 ± 12.7	62.5 ± 7.6	55.9 ± 2.0 *	0.029	0.187
normalized FL	0.36 ± 0.48	0.18 ± 0.03	0.16 ± 0.01 *	0.015	0.237
muscle thickness (mm)	16.8 ± 1.5	17.3 ± 2.4	18.9 ± 1.9	0.065	0.183
pennation angle (°)	16.2 ± 1.9	18.9 ± 2.8	19.3 ± 2.4 *	0.008	0.280
L*_AT_* (cm)	20.43 ± 2.27	20.23 ± 1.73	21.57 ± 2.41	0.413	0.063
normalized L*_AT_*	0.61 ± 0.05	0.60 ± 0.07	0.62 ± 0.07	0.726	0.023
CSA (mm^2^)	55.82 ± 11.52	56.13 ± 8.29	51.95 ± 6.95	0.387	0.068
T*_MAX_* (N∙m)	106.55 ± 31.07	93.88 ± 22.50	96.62 ± 27.35	0.418	0.063
normalized T*_MAX_* (Nm∙kg^−1^)	1.56 ± 0.039	1.35 ± 0.30	1.41 ± 0.34	0.345	0.076

Note: * significant difference between FFS and NR (*p* < 0.05); FFS: forefoot strikers; RFS: rearfoot strikers; NR: nonrunners; FL: fascicle length; MT: muscle thickness; PA: pennation angle; L*_AT_*: rest length of AT; CSA: cross-sectional area of AT; T*_MAX_*: peak plantar flexion torque.

## 4. Discussion

The effects of different long-term habitual foot strike patterns on gMTU morphology and ankle plantar flexion torque were evaluated in this study. Consistent with our hypotheses, the FFS runners had both a longer FL (*p* < 0.05) and longer normalized FL (*p* < 0.05) than the NRs. In contrast, the PA of the FFS runners was smaller than that of the NRs. These findings may be explained by the FFS runners having more eccentric mechanical stimulations the during early stance phase to lengthen muscle and a more efficient force transmission from the muscle to the tendon. However, the L*_AT_*, CSA of the AT, and plantar flexion torque did not differ between the three groups.

In the present study, a difference in the FL was found between the FFS runner group and the NR group. The longer FL of the FFS runners might be attributed to an eccentric mechanical stimulation that lengthens the muscle during the early stance phase of running [31]. This inference is supported by a previous study that reported an increased FL after eccentric exercises [26]. The longer the FL, the greater the potential maximal velocity of shortening, allowing the muscle to more rapidly contract while running [32,33]. In addition, a shorter FL would have a reduced working range limitation during running compared with a longer FL as a result of the fewer sarcomeres in the series [34,35]. Accordingly, the longer FL of the FFS runners may also be a long-term adaptive change that might be protecting the MG from strain under repetitive high loads and acute stretching in the early stance phase during running. However, there was no significant difference found in the FL and normalized FL between the RFS runners and NRs, despite the tendency for RFS runners, along with NRs, to have a longer FL and normalized FL. This phenomenon could be explained by the features of the RFS. The RFS have a longer stance phase, and compared with that among the FFS group, the MG of the runners experienced fewer eccentric contractions in the early stance phase [36].

In this study, the FFS runners were found to have a smaller PA than the NRs. Machado reported that the PA was negatively correlated with the metabolic cost of running in long-distance runners [37]. Therefore, a smaller PA within a certain range could transmit a greater force to the tendon. However, unexpectedly, there were no significant adaptive changes observed in the CSA of the AT after long-term running among the FFS runners. This finding could be attributed to the fact that there are fewer blood vessels in the AT, indicating a slower metabolic rate [38]. Similarly, Hansen et al. observed no significant difference in the CSA of the AT after long-term running [39]. Therefore, FFS runners may have a more efficient force transmission, with more energy cost in the propulsion phase in terms of muscle morphology.

In the comparison of the MT, no significant difference was found between the FFS runners, RFS runners, and NRs. Our results agree with previous findings on long-distance runners [16]. This finding indicates that the long-distance habitual foot strike patterns had little effect on the MT of medial gastrocnemius [40]. Our study provides some morphological support to the findings among long-distance runners who tend to have slender limbs, especially for the lower extremities [41]. Therefore, the unchanged MT, which was positively correlated with MVICs, could also be explained by the finding that there was no significant difference in the ankle plantar flexion torque between the three groups [42,43].

In terms of the strike pattern, there was no difference observed in the morphology or plantar flexion torque between the FFS and RFS runners in this study, which is consistent with previous findings. Gonzales et al. [44] investigated the lower limb muscle morphology of long-distance FFS and RFS runners and found no significant difference in the FL or PA of the MG. While they observed that FFS runners had a greater plantar flexion torque than RFS runners, in their study, the weekly distance run by the FFS runners was much greater than that of RFS runners. In the present study, even though there were no statistically significant differences in ankle plantar flexion torque between the FFS and RFS groups, the tendency of a larger ankle plantar flexion torque in FFS runners along with RFS runners was still worthy of consideration. However, the significant differences found in the morphological features between the FFS and NR runners in the present study indicate that RFS runners have less stimuli for inducing significant adaptive changes in the gMTU at the morphological level and ankle plantarflexion torque during long-distance running. Another explanation for the lack of significant difference in the mechanical function between the three groups may be the contradictory functions of PA and FL. When the FL increased, the pennation angle decreased, and the MG generated less force [45]. The contradictory results on the outcomes of habitual foot strike patterns may be influenced by an inconsistent amount of running.

The current study has some limitations. Only male participants were recruited, so gender difference should be considered in future studies. The gMTU was the sole target muscle tendon unit used to compare individuals with different running habits; additional investigation on other target muscles, such as the lateral gastrocnemius and soleus, should be performed. Moreover, only the static morphological features of the muscles were evaluated. Future studies should consider the dynamic morphological properties of the muscles to obtain more information on running outcomes.

## 5. Conclusions

The FFS group had a longer FL and a smaller PA of the MG than the NR group. However, no other significant difference was observed between the RFS and other groups. The longer FL suggests that the maximal shortening velocity of the MG is greater, and the possibility of MG strain is lower, which may be an adaptive protective mechanism that develops after long-term running among FFS runners. The smaller PA of the MG also indicates a more efficient transmitting force among these runners. Therefore, long-term running with an FFS pattern could improve the efficiency of the transmitting force and protect the MG from strain under repetitive high loads. The findings of the present study provide more morphological evidence for future study and the application of FFS running training.

## Figures and Tables

**Figure 1 bioengineering-10-00264-f001:**
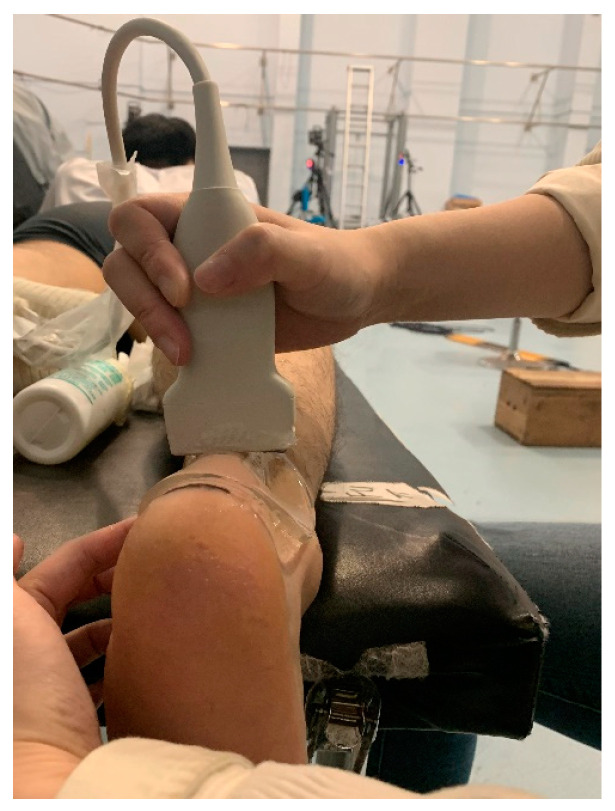
The measurement location of AT’s CSA.

**Figure 2 bioengineering-10-00264-f002:**
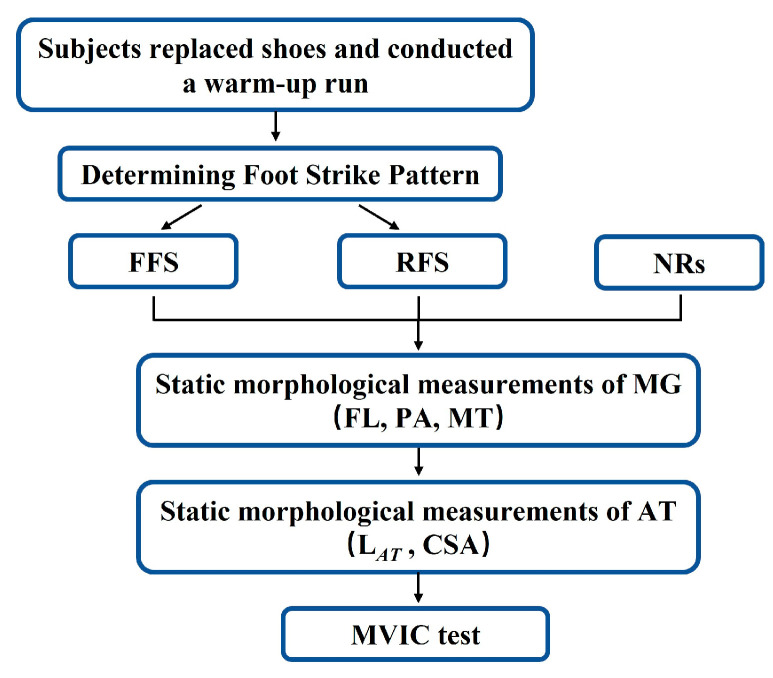
Diagram of the experimental flow chart.

**Figure 4 bioengineering-10-00264-f004:**
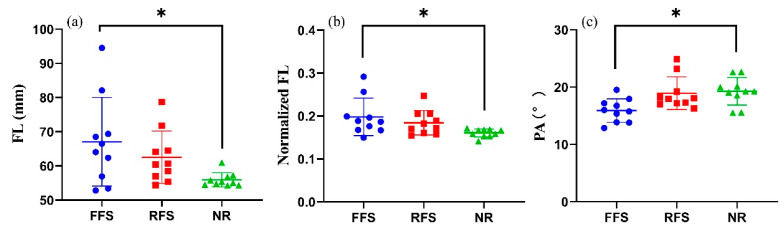
Effects of habitual foot strike patterns on the FL (**a**), normalized FL (**b**), and PA (**c**) of MG. Note: * *p* < 0.05. FFS: habitual forefoot strikers; RFS: habitual rearfoot strikers; NR: nonrunners; FL: fascicle length; normalized FL: FL divided by the shank length; PA: pennation angle.

**Table 1 bioengineering-10-00264-t001:** Basic information of the participants.

Group	Age (Years)	Height (cm)	Weight (kg)	Weekly Volume (km)	Running Years
FFS (n = 10)	27.50 ± 8.68	175.75 ± 7.69	69.01 ± 6.96	39.50 ± 18.92	5.40 ± 3.47
RFS (n = 10)	30.10 ± 5.07	173.00 ± 5.16	69.59 ± 10.11	34.60 ± 13.72	4.00 ± 1.70
NR (n = 10)	25.40 ± 1.78	175.00 ± 4.74	68.01 ± 6.48	/	/
*p* value	0.147	0.675	0.910	0.516	0.272

## Data Availability

Data available upon request.

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
