# Peer review of "The Effects of Habitual Foot Strike Patterns on the Morphology and Mechanical Function of the Medial Gastrocnemius–Achilles Tendon Unit"

_bioengineering, 2023, doi:10.3390/bioengineering10020264_

Round 1

Reviewer 1 Report

The authors compared gastrocnemius medialis morphology in 30 male runners with fore, rearfoot striking pattern and non-runners. They found longer fascicles and smaller pennation angles in forefoot striking runners.

The strong part of the study is that it is an interesting observation with a sound interpretation. The weak part was that some of the methods to assess morphologies may need to defined in greater detail.

Introduction:

Line 15: the authors did not investigate the function of the muscles during locomotion I would replace that with morphology.

line 23: the authors may consider testing only normalized FL , the absolute may just be presented as values, it does not make sense to compare subjects muscle length of different bodyheights.

In the last paragraph I would rather focus on the muscle strain, such as done in the conclusion at the end of the paper, other aspects are too much speculations also about the transfer of forces.

Line 45: papers 11 and 12 are about hamstrings, not the calf muscles.

Line 48: midfoot strike is also a typical pattern.

Line 47: there is not only academic interest in foot strike patterns, it has great relevant for shoes industries and sports and injuries.

Line 51: the phase of the plantarflexion angle need to be defined I would rather guess that the peak doriflexion angle and the peak ankle plantarflexion moment in stance is greater.    

Line 60: why did the explanation appear in the next paragraph, this line shift seems not appropriate here.    

Last paragraph: how were all these hypotheses based on the existing literature?

Methods:

Line 76: the literature refers to the elderly.  CSA did not appear in the abstract, I would rather do a power analysis on the main outcome parameter.

Line 77: why were only male runners investigated? nowadays with the gender correctness this can be very problematic and papers may be rejected when there is not a plausible reason behind this selection.

Line 94: which frequency was set and how long was the scanner. I doubt that with the scanner fit 9 cm fascicles so that a lot of extrapolation must be done.

What was the ankle angle at which FL and PA was measured?

At which orientation medio-lateral was the probe aligned to measure the thickness?

Line 135: how was shank length defined?

Line 145: how was the attachement point of the AT to the calcaneus defined, it is rather a broad area.

Results

Figure 4: over 9 cm is an extremely large FL where a lot of interpolation must be done and fascicles are not uniform in particular when inserting to the MT junction.

Table 2: the normalized FL in FFS is larger than in the diagram, the digits need to be revised the normalized units have a lower accuracy they may be presented in percent. Two digits after the decimal point are too accurate considering the errors of ultrasonography.

Line 203: the longer the fascicles the smaller the PA in the same volume. A lower PA has also a lower force since less fascicles can be placed in parallel. This counterargues the transfer to the aponeuroses, I assume you mean here the cos(PA) relation frequently used in the hill muscle model to reduce the force applied along the line of muscle transfer.

Reviewer 2 Report

In this study, author analyse the morphology of the medial gastrocnemius, Achilles tendon and ankle torque during maximal voluntary isometric contraction for three groups of male subjects. Long distance runners with forefoot and rearfoot strike patterns configured two of these groups, whereas non-runners were the control group. Regarding the morphological measurements, the authors find with statistical significance that runners with a forefoot strike pattern have longer fascicle length and pennation angle than the control group. 

The paper is well written, structured and the results provide new insights in the running biomechanics. However, I find some issues that should be improved when authors discuss their findings. For example, I couldn’t find a comparison of pennation angles with other values in the literature. Machado et al 2022 reported for the medial gastrocnemius 24.36 +/- 3.32 º pennation angles for 17 recreation runners. What could cause this difference? Same could be applied for other parameters.  

Some other comments:

MATERIALS AND METHODS

2.1 Participants 

Line 76: Please provide an explanation for parameters f, alpha and beta.

2.3 Procedures

Figure 2. At first sight, the diagram seems confusing for me. Why showing the morphological measurements of MG and AT in “parallel”? 

2.4 Data Analysis

Line 146: The reference to Figure 3,b here is misplaced, no description of LAT or shank length appear in it.

DISCUSSION

Line 190: Cite [29] appears as a superscript. 

Line 200-211. According to the negative correlation of the PA with respect to the running metabolic cost (Machado et al 2022), the last sentence of this paragraph is contradictory. According to your results FFS runners develop a lower PA which represents a higher metabolic cost. Could authors clarify this?

Reviewer 3 Report

General Comments

Congratulations for this study. However, authors should better express the question

in this study that will be answered. I didn’t  identify it... Please, see the specific comments.

Specific Comments

Abstract

Line 16 to 21 – Authors should emphasise the experimental protocol. This information was almost scarce in this section.

Line 21 to 26 – Insert mean, standard deviation and effect size values. The p value is insufficient to show the relevance of the results.

Line 26-29 – Could authors expose some practical application for athletes and coaches?

Introduction

Line 39 to 43 – Reduce these three sentences in one.

Why authors expose information regarding the kinematic patterns of running? Did the authors measure these variables? Why authors did not expose information regarding the evidences on variables tested by ultrasonography method?

What is the question of this study that will be answered?

Methods

Authors should insert the errors of measurements.

What kind of references authors used to describe the data analysis?  Please quote at least three references that use this type of analysis. 

Results

The results are repetitive on text and tables. Please correct it.

Discussion

In the 1st para authors should insert a brief explanation regarding the main findings.  

Conclusion

Where is the practical application of this study?

References

In their reference list, the name of the journals are not patronized.     

Round 2

Reviewer 2 Report

All my comments have been considered in the new version of the manuscript.

Author Response

Thank you for your recognition. Your valuable suggestions and comments helped us improve our manuscript a lot.

Reviewer 3 Report

Specific Comments

Introduction

Line 38 to 40 – The first two sentences are not connected.

Line 71 to 76 – Rewrite all these sentences. It is unclear to the reader. 

Methods

Authors should insert the errors of measurements.

What kind of references authors used to describe the data analysis?  Please quote at least three references that use this type of analysis. 

Results

The results are repetitive on text and tables. Please correct it. See for example lines 169-170.
